# Understanding the Role of Connexins in Hepatocellular Carcinoma: Molecular and Prognostic Implications

**DOI:** 10.3390/cancers16081533

**Published:** 2024-04-17

**Authors:** Stavros P. Papadakos, Elena Chatzikalil, Konstantinos Arvanitakis, Georgios Vakadaris, Ioanna E. Stergiou, Maria-Loukia Koutsompina, Alexandra Argyrou, Vasileios Lekakis, Ippokratis Konstantinidis, Georgios Germanidis, Stamatios Theocharis

**Affiliations:** 1First Department of Pathology, Medical School, National and Kapodistrian University of Athens, 11527 Athens, Greece; stpap@med.uoa.gr (S.P.P.); ehatzikali@gmail.com (E.C.); 2Division of Gastroenterology and Hepatology, First Department of Internal Medicine, AHEPA University Hospital, Aristotle University of Thessaloniki, 54636 Thessaloniki, Greece; arvanitak@auth.gr (K.A.); vakadarisgeorgios@gmail.com (G.V.); 3Basic and Translational Research Unit, Special Unit for Biomedical Research and Education, School of Medicine, Faculty of Health Sciences, Aristotle University of Thessaloniki, 54636 Thessaloniki, Greece; 4Pathophysiology Department, School of Medicine, National and Kapodistrian University of Athens, 11527 Athens, Greece; stergiouioa@med.uoa.gr (I.E.S.); marilikoutsomp@gmail.com (M.-L.K.); 5Academic Department of Gastroenterology, Laikon General Hospital, Athens University Medical School, 11527 Athens, Greece; argyalex89@gmail.com (A.A.); lekakis.vas@gmail.com (V.L.); 6Department of Internal Medicine, University of Connecticut, Farmington, CT 06030, USA; konstantinidis@uchc.edu

**Keywords:** connexins, gap junctions, hepatocellular carcinoma, expression, chemosensitivity

## Abstract

**Simple Summary:**

Hepatocellular carcinoma represents the majority of all liver cancers and the fourth cause of cancer-related deaths worldwide. Despite the progress in diagnostic and therapeutic evaluation, its prognosis remains dismal, with mortality rates increasing per year. As the investigation of the human genome project advances, so does the deeper understanding of tumor biology, with numerous molecular biomarkers proved to play a promising role in clinical practice, in terms of diagnosis, surveillance, and the prediction of treatment efficacy. One important aspect is evaluating the potential role of new ancillary biomarkers associated with tumor invasiveness and response to available therapeutic options, which is useful in treatment monitoring and may reduce the health costs faced by standard methods.

**Abstract:**

Connexins, a family of tetraspan membrane proteins forming intercellular channels localized in gap junctions, play a pivotal role at the different stages of tumor progression presenting both pro- and anti-tumorigenic effects. Considering the potential role of connexins as tumor suppressors through multiple channel-independent mechanisms, their loss of expression may be associated with tumorigenic activity, while it is hypothesized that connexins favor the clonal expansion of tumor cells and promote cell migration, invasion, and proliferation, affecting metastasis and chemoresistance in some cases. Hepatocellular carcinoma (HCC), characterized by unfavorable prognosis and limited responsiveness to current therapeutic strategies, has been linked to gap junction proteins as tumorigenic factors with prognostic value. Notably, several members of connexins have emerged as promising markers for assessing the progression and aggressiveness of HCC, as well as the chemosensitivity and radiosensitivity of hepatocellular tumor cells. Our review sheds light on the multifaceted role of connexins in HCC pathogenesis, offering valuable insights on recent advances in determining their prognostic and therapeutic potential.

## 1. Introduction

Liver cancer represents a formidable global health challenge, with its prevalence steadily rising on a worldwide scale [1]. Internationally, it is ranked as the sixth most frequently diagnosed cancer and the fourth leading cause of cancer-related mortality [2]. Among its various forms, hepatocellular carcinoma (HCC) stands as the most prevalent one, constituting the majority of primary liver malignancies [3]. Gender-based disparities are evident, as men exhibit an incidence rate from two- to four-times higher than those of women across most countries [4]. The global distribution of HCC correlates with age of acquisition and with viral hepatitis within the population. In regions with high incidence, hepatitis B virus (HBV) transmission during birth (perinatal transmission) predominates, resulting in HCC diagnosis occurring approximately a decade earlier as compared with regions like North America and Europe where the median age of diagnosis is in the seventh decade of life [5]. 

HCC emerges as a consequent of prolonged and sustained inflammation [6]. This process can be instigated by intrinsic factors, such as inherited or acquired genetic mutations, or by extrinsic risk factors. Chronic infection with hepatitis C virus (HCV) and HBV, and metabolic dysfunction-associated steatotic liver disease (MASLD), formerly known as non-alcoholic fatty liver disease/non-alcoholic steatohepatitis (NAFLD/NASH), stand as the primary etiological factors fostering the development of liver cirrhosis, which subsequently predisposes to HCC development [7,8]. These factors assume pivotal roles in HCC progression by inducing molecular and structural changes in mature hepatocytes or stem cells, initiating tumor growth [9,10]. Diagnosis of HCC is typically based on non-invasive criteria, although there is an increasing demand for molecular profiling of the tumor via tissue biopsies in clinical settings [11]. Treatment strategies are guided by tumor staging aligning with the Barcelona Clinic Liver Cancer (BCLC) staging system, with resection, transplantation, and local ablation being favored for patients with early stage HCC tumors [12]. On the contrary, transarterial chemoembolization (TACE) and systemic therapy are preferred for intermediate- and advanced-stage tumors, respectively [13]. 

Despite scientific progress, dismal prognosis persists for the majority of patients diagnosed with HCC [14]. Connexins (Cx) represent a protein family integral to the formation of intercellular channels found within gap junction (GJ) plaques, as well as within single transmembrane channels known as hemichannels [15]. GJs are implicated in a spectrum of physiological processes crucial for cellular homeostasis, encompassing cell differentiation [16], angiogenesis [17], and stem cell maturation [18]. Hemichannels assume a vital function in autocrine and paracrine signaling pathways [16]. Within the healthy liver, hepatocytes predominantly express Cx32 and Cx26, while non-parenchymal cells like Kupffer cells mostly express Cx43 [19]. In HCC tissue, Cx43 expression increases [20], while the expression of Cx26 and particularly Cx32 shows a decrease [21]. Furthermore, recent studies have also indicated the potential role of Cx expression in cancer therapeutics. Increased Cx26 expression can play an important role in modulating radiosensitivity in squamous cell carcinoma [22], while impaired GJs have been correlated with oxaliplatin chemosensitivity among patients with HCC [23]. 

Our review delves into the pathophysiological role of connexins in HCC, aiming to provide readers with a deeper understanding and bridge potential knowledge gaps regarding the involvement of connexins in HCC, offering valuable insight for its future management and facilitating the emergence of innovative treatment approaches.

## 2. Connexin Structure, Regulation, and Involvement in Tumorigenic Processes

### 2.1. Structure and Formation of Gap Junctions and Connexins 

The development of multicellular organisms, homeostasis, and growth control require coordinated cellular communication [24]. Direct intercellular communication, as well as communication between intracellular and extracellular environments, is favored by protein families which enable interactions between adjacent cells, partially regulating vital processes, such as the rapid spread of action potentials, the exchange of metabolism byproducts, and the diffusion of nonorganic ions, secondary messengers, and other small water-soluble molecules [25,26,27]. 

An important mode of cell–cell interaction is created by a direct conduit between the cytoplasm of adjacent cells, which is favored by the presence of gap junction (GJs) channels [28]. GJs are the only known cellular structures that allow a cell-to-cell direct transfer of signaling molecules bridging the opposing membranes of neighboring cells with hydrophilic channels [28,29]. GJs channels consist of two hemi-channels, the connexons, each one belonging to the respective one of the two adjacent cells, a formation which leaves a 2–4 nm “gap” between the adjacent cell membranes [29,30]. In this “gap”, two connexons dock and form a tightly sealed double membrane intercellular channel [29,30]. The central pore of the intracellular channel stands as an electrical and biochemical coupling “path” that links the cytoplasm of two cells enabling the exchange of small signaling molecules (up to 1.4 nm) with relatively low specificity, including ions (K^+^, Ca^2+^) second messengers (cAMP, cGMP, inositol triphosphate (IP_3_)), small metabolites (glucose), antigens, and microRNAs [30]. Notably, hemichannels can also work as uncoupled channels, allowing the exchange of chemical information between the cytoplasm and the extracellular environment [30].

Intracellular communication is favored by special structures in many kinds of organisms, from plants using structures called “plasmodesmata”, to vertebrates and invertebrates, which use gap junctions as previously described [28,31]. In humans and chordate animals, GJ protein structures are formed by integral multi-pass transmembrane protein subunits called “connexins” that are encoded by a family of genes also termed “connexins”, identified with a symbol starting with “*GJ*” (for gap junction) and categorized into different subfamilies, mainly regarding their gene structure, gene homology, and specific sequence motifs [28,30]. Currently, five connexin subfamilies are recognized (α, β, γ, δ, and ε or *GJA*, *GJB*, *GJC*, *GJD*, and *GJE*), and 21 distinct *connexin* genes have been identified in humans [32]. Regarding connexin proteins nomenclature, two modes are met in the literature. The first one, which is most commonly used, depends on the molecular mass in kilodaltons and defines connexins as follows; Cx26 represents the connexin with 26kDa molecular mass; Cx32 represents the connexin with 32kDa molecular mass, Cx43 represents the connexin with 43kDa molecular mass, etc., while the other one uses alphabetic characters based on evolutionary considerations (e.g., *GJB2* stands for gap junction beta 2, referring to Cx26; *GJA3* stands for gap junction alpha 3, referring to Cx46) [30,32]. 

Connexins have a common 3D structure consisting of four hydrophobic transmembrane domains (TM1-TM4), one intracellular (cytoplasmic) loop (CL) and two extracellular loops (E1-E2), and N- and C- terminal cytoplasmic domains [29,30]. The transmembrane domains, the extracellular loops, and the N-terminal cytoplasmic domain are highly conserved in humans. The cytoplasmic loop varies in terms of length, which differs between the different connexin subfamilies, and is sensitive to various post-translational modifications, while the C-terminal domain presents with great variations in terms of sequence and length [33]. The transmembrane domains constitute the wall of the channels and are connected by an intracellular loop and the extracellular loops [28,30]. The extracellular loops, each one containing three cysteine regions which form a disulfide bond, are involved in adjacent cell recognition and docking [28,30]. The N-terminal cytoplasmic domain is involved in the connexin oligomerization into hemichannels and the ensuing connexin trafficking, while it enables selective interactive and docking processes via a selectivity signal [30]. The C-terminal cytoplasmic domain is involved in the phosphorylation of several connexins (Cx31, Cx32, Cx37, Cx40, Cx43, Cx45, Cx47, and Cx50), thus regulating the assembly and modulation of the channels’ physiological properties [30]. It also plays a crucial role in the oligomerization processes and in the flow regulation of the intercellular Ca^2+^ by binding calmodulin, affecting the biochemical and electrical properties of hemichannels [30]. Interestingly, the connexin compartments presenting variations between family members, the cytoplasmic loop, and the C-terminal cytoplasmic domain can bind to a variety of structural proteins (e.g., zonulin 1 and 2, claudin 1), resulting in the formation of advanced, in terms of stability and functional capacity, protein complexes [30,34].

Connexin biosynthesis starts with their contranslational integration into the endoplasmic reticulum membranes, where they are found as single units [35]. Their oligomerization into hexameric hemichannels (connexons) occurs during their delivery to the Golgi network that ensues [29]. Hemichannels consist of either a single or more types of connexins, termed homomeric or heteromeric, respectively, and they either form homotypic channels, or heterotypic channels with different homomeric or heteromeric hemichannels [29,30]. The hemichannels are then delivered to cell membranes through microtubules and cluster into temporally organized gap junction channel aggregates, termed plaques, removing the old channels from the center of the plaque while being added to the periphery [35]. The old channels, after being removed, are either degraded by lysosomal enzymes or are targeted at the proteasomal pathway [35]. Not all hemichannels are transported to cell-to-cell contact areas; some of them are found at unopposed areas of the cell membrane, but they can also be occasionally functional, mainly in stress conditions, in which they enable the release of adenosine triphosphate (ATP), glutamate, and nicotinamide adenine dinucleotide ions (NAD+) into extracellular space [35].

### 2.2. Regulation of Connexins Expression 

Positive regulation of gap junctions is a result of the activity of many intracellular proteins and cytokines [36]. In more detail, their synthesis in the endoplasmic reticulum and their transportation to the Golgi apparatus are considered to be promoted by the TGF-β2/Smad3 pathway and by their interaction with CIP75 (connexin-interacting protein of 75 kDa) [37]. The subunits of microtubules and actin filaments, α/β-tubulin and F-actin, positively regulate connexin transportation via microtubules’ stabilization, which is important for the maintenance of cell polarity, while the binding protein drebrin is required as a mediator of F-actin and connexin binding, positively regulating the transportation process [38,39]. Connexin transportation is also enhanced by the zonula occludens-1 (ZO-1) protein, which mediates hemichannel assembly, presenting a positive correlation between its expression and the number of GJ plaques, thus increasing the transport capacity of substances [40]. Regulation of the GJ gate opening is also of particular importance and is mainly achieved via cAMP/protein kinase A (PKA) signaling, which is involved in promoting the connexin modification for gate opening [41]. Moreover, the stabilization of GJs and their expression levels’ maintenance are mostly regulated by signaling pathways; characteristic examples are the stabilization of GJs via β-catenin activity and the enhancement of connexin expression via the Wnt/β-catenin signaling pathway [42].

Regarding GJs’ negative regulation, it is considered a result of proteasome activity, which acts in a ubiquitin-dependent manner (as modulated by CIP75 after connexin synthesis), leading to connexin degradation [43]. A direct degradation is also achieved via lysosomes or autophagy, with or without entering the early and late endosomes, before finally being degraded [44,45]. Epidermal growth factor (EGFR) is also a negative regulator of connexin activity, promoting endocytosis through the MAPK and PKC signaling pathways [46]. Moreover, modification of connexins’ tyrosine sites by Src prevents connexins from binding to tubulin, while connexin downregulation increases the focal adhesion kinase (FAK)-Src activation [47]. Interestingly, c-Src regulation of epithelial-mesenchymal transition (EMT) via the PI3K/Akt pathway may regulate connexin activity, as the decreased binding ability of c-Src to connexin members increases the activity of Akt, enhancing the invasive and metastatic capacity [47].

### 2.3. Connexins in Neoplastic Processes 

GJ proteins regulate many tumorigenic processes, due to their structural characteristics, or by triggering mechanisms that enhance neoplastic development. Regarding their structural acquired characteristics, it is worth highlighting that maintaining cell polarity is important for cell stability and physiologic intracellular function, while its dysregulation is associated with neoplastic activity [48,49]. The disruption of GJs cause the loss of apical polarity, microtubules polarization, and dysregulation of epithelial cell homeostatic mechanisms in many types of cancer, simultaneously modifying telomerase activity, resulting in cell de-differentiation and immortality [50,51]. Secondly, it has been demonstrated that GJ intracellular channels promote transendothelial cell metastasis and tumor angiogenesis, thus being targets for anti-tumor therapy [52]. Moreover, interacting with cell cycle regulatory molecules (e.g., promoting P21cip1 and P27kip1 expression, regulating Ca^2+^ transmission) and presenting a positive correlation between their expression levels and the G1 phase duration, GJ proteins are involved in cell cycle regulation by delivering either pro-death or anti-death factors [53,54]. Regarding cancer metabolism, GJs are considered to be pathways transporting hypoxic cell metabolites, such as lactate into oxygen-rich cells and, simultaneously, HCO_3−_ from oxygen-rich cells to hypoxic cells, while it is also involved in the dysregulation of glycolysis [55]. It is also worth mentioning that GJ proteins are involved in EMT processes either as monomers or in a GJIC-dependent manner, regulating the loss of cell polarity and cell-to-cell contact [52,56]. As a result of the connexin dysregulation of expression, the adhesive ability between cancer cells is disrupted, rendering them more capable of invading surrounding tissues across the basement membrane and to metastasize in distant sites [57].

The structure of connexins, alongside positive and negative regulatory mechanisms and their involvement in tumorigenesis, are illustrated in Figure 1. 

## 3. Defining Connexins’ Expression and Functionality as Potential Prognostic and Therapeutic Markers in HCC

### 3.1. Connexin 32 (Cx32)

#### 3.1.1. Connexin 32 Expression

Cx32 is the major connexin isoform of hepatic GJs and is broadly expressed in nearly 90%, while in terms of its expression, tumorigenesis regulation, and its potential effects on chemosensitivity, it is the most widely studied [58,59]. Considering Cx32 genes’ tumor-suppressing effects and their involvement in various processes (tumorigenesis, cell proliferation, apoptosis, invasion, metastasis) [60,61], several studies have been conducted investigating the association between Cx32 expression patterns and established mechanisms related to carcinogenesis (e.g., EMT, PI3K/Akt pathway, necroptosis/apoptosis), in order to establish its potential use as a diagnostic and prognostic marker in HCC [59,62,63,64,65,66,67,68,69,70,71,72,73,74]. However, further in vivo investigation is needed to draw robust conclusions for its use in clinical practice. 

Cx32 expression has been studied in vitro, using cell lines and HCC tissue samples, and in vivo, utilizing mouse models, demonstrating a characteristic expression pattern in HCC cells, associated with various tumor-related factors [59,62,63,64,65,66,67,68,69,70,71,72,73,74]. As a general rule, Cx32 presents with lower expression in HCC as compared to normal, even para-cancerous, liver tissue, as observed in both mRNA and at the protein level, and it is characterized by an ectopic expression from the cell membrane (where it is normally mainly expressed) to the cytoplasm of cancerous liver cells, especially in advanced malignant and invasive HCC [62,66,68,69]. Cx32 expression levels in HCC vary regarding tumor characteristics, being negatively correlated with histological grade and lymph node (LN) metastasis; however, no correlation between Cx32 expression and age, gender, tumor size, TNM stage, history of liver disease, vascular embolus, chronic HBV or HCV carriage, tumor necrosis, or tumor hemorrhage has been established [66]. In addition to HCC cells, a downregulation of Cx32 expression in cancer stem cells (CSCs) has been observed, which is extensively described herein [70]. 

#### 3.1.2. The Connexin 32 Role in Tumorigenesis and Metastasis in HCC

Cx32 is considered to play a role in HCC’s proliferative, invasive, and metastatic potential. This could partially be explained by the insufficient quantity of GJs in HCC tissues; the significant reduction of Cx32 that has been observed in the S-phase of cell-cycle suggests that quantitative changes in GJ expression are associated with the control mechanisms of cell’s proliferation [63]. HCC tissues present a higher proliferation rate than normal liver tissue, which leads to a decrease amount of GJs, while it is suggested that pre-neoplastic cells with a reduced number of GJs might have an increased capability of proliferating, and thus, a higher possibility for HCC development [75]. Regarding the Cx32′s expression characteristics, the cytoplasmic accumulation of Cx32 on tumor cell populations has been identified as the main mechanism that is involved in tumor invasion and metastasis. Some studies have suggested that abnormal localization of Cx32 is more important for the tumorigenic activity as compared with translational dysregulation [63,76]. Ectopic overexpression of Cx32 in the cytoplasm is considered to increase the proliferation, motility, and invasiveness of HCC cells, as well as to expand CSC population development, elevating CSC renewal rate and increasing sphere formation [64,65]. These effects have been recently investigated in a few studies described below.

Li et al. were the first to investigate Cx32 cytoplasmic accumulation as a potential mechanism for migration and metastasis enhancement, using HuH7 and Li-7 HCC cell lines with overexpressed Cx32 by doxycycline withdrawal (Tet-off cells) [64,77]. In HuH7 Tet-off Cx32 cells, Cx32 was mildly expressed in the cytoplasm when cultured in doxycycline supplemented medium, while in doxycycline-free medium, Cx32 expression occurred mainly in the cytoplasm but not in cell-to-cell contact areas, suggesting that the absence of GJ channels from HCC occurs due to impaired intracellular internalization of Cx32 to the cell membrane and not due to insufficient quantity of Cx32 protein. Moreover, the overexpression of cytoplasmic Cx32 protein was associated with enhanced proliferation and motility, as well as higher invasive capacity of HuH7 Tet-off Cx32 cells in a doxycycline-free medium. Similar observations were obtained when conducting the same experiments using Li-7 HCC cells lines. These results highlight that Cx32 overexpression in the cytoplasm is associated with malignant tumor phenotypes and leads to tumor progression via several mechanisms (invasion, proliferation, and enhanced motility) [64].

In the same context, a recent study by Kawasaki et al. focused on investigating the Cx32 expression involvement in CSCs metastatic-related processes [65]. CSCs are considered as a distinct population which offers support to the tumor cell populations, presenting a self-renewal ability and determining the incidence of metastasis [78,79,80]. Based on the study conducted by Kawasaki et al., who investigated the effects of cytoplasmic Cx32 protein on the regulation of CSC population using a Tet-off system (where overexpression of Cx32 was induced by withdrawal of doxycycline from the cell culture medium) [77], the cytoplasmic Cx32-mediated potentiation of metastasis is considered to involve expansion in the CSC population [65]. Major observations of Kawasaki’s et al. study were based on the functionality of a small distinct population of CSCs termed “the side population (SP)”, which is the fraction into which stem cells in normal tissues are efficiently enriched, considered as a CSC marker [65,81,82]. A 10-times-higher proportion of SP fraction was observed in doxycycline medium cultures, while this difference was not observed in HuH7 Tet-off mock cells, indicating that cells with a higher Cx32 cytoplasmic expression contain larger SP fractions, possibly due to CSCs’ self-renewal enhancement. Moreover, pre-tumor self-renewal spheres of SP cells from HuH7 Tet-off Cx32 cells were increased in both number and size in a doxycycline-free medium compared with their counterparts in a doxycycline-supplemented one. In contrast with this, SP cells from HuH7 Tet-off mock cells were able to form large spheres regardless of whether they are doxycycline or free-medium cultured. These results suggest that cytoplasmic accumulation of Cx32 protein enhances self-renewal of CSCs in HuH7 cell line [65]. 

#### 3.1.3. Exploring the Interplay between Connexin 32, EMT Signaling, and Chemoresistance in HCC

EMT is a transient and reversible switch from a polarized epithelial to a fibroblast or mesenchymal cellular phenotype (with decreased intercellular adhesion and increased cell motility and invasive capacity) [83,84,85,86,87,88]. EMT is a central event in early stages of embryonic development; presents a pivotal role in tumorigenesis in many types of tumors, including HCC; and is associated with acquired drug resistance for the cells being at the mesenchymal differentiation state [83,84,85,86,87,88]. It is considered that cells which have undergone EMT slightly express or lose epithelial markers (e.g., E-cadherin), while they also highly express mesenchymal markers (e.g., Vimentin), with a parallel upregulated expression of transcription factors (e.g., Snail) [66,89]. Currently, there are three different studies aiming to propose a correlation between Cx32 expression, the EMT signaling network, and chemoresistance in oxaliplatin and doxorubicin. 

Yang et al. were the first to investigate a correlation between Cx32 expression, EMT and HCC response to oxaliplatin chemotherapy, conducting two different studies within a 2-year interval [59,66]. In their first study, they used tissue samples (76 HCC samples and 20 normal controls), cell lines (normal hepatic cell line: LO2, HCC carcinoma cell lines: HepG2, Huh7, and SMMC-77210), and mouse models (Huh7-hCx32 cells or Huh7-vec cells inoculated) to study Cx32 and EMT markers expression [66]. Downregulating Cx32 expression in SMMC-77210 cells, which normally present a high Cx32 expression, and upregulating Cx32 expression in HepG2 cells, which normally present a low Cx32 expression, resulted in different morphological changes: in the first case, the cells obtained an epithelial phenotype, with a decreased E-cadherin and an increased Vimentin and Snail expression, while in the second case, the cells obtained a mesenchymal phenotype, with an increased E-cadherin and a decreased Vimentin and Snail expression. Snail is a zinc-finger transcriptional repressor, which is considered as a potential oncogene in various tumors, including HCC [90], enhancing EMT and promoting metastasis [91,92]. Using similar techniques, a potential effect of Snail expression was shown in EMT-associated invasion in HCC cells, while the expression of phosphorylated β-catenin (Y654), the presence of which marks transcriptional activity and nuclear translocation of β-catenin [93], was negatively correlated with Cx32 expression, accompanied with a consistent change in Wnt1 expression. When an inhibitor of the Wnt signaling pathway was used, the EMT changes, resulting from Cx32 silencing in SMMC-7721 cells and annulling Snail upregulation due to Cx32 downregulation, were reversed. It was, overall, suggested that Cx32 downregulation upregulates Snail expression and promotes EMT via Wnt1 signaling pathway activation. The Cx32/β-catenin/Snail pathway was proposed as a potential therapeutic target in advanced HCC, which is recurrent to established chemotherapy with oxaliplatin [66].

In their second study, Yang et al. used HepG2, Huh7, and SMMC-7721 HCC cell lines, developing oxaliplatin-resistant cell lines via increasing the exposure to oxaliplatin in several time intervals, utilizing parental HCC as controls, aiming to associate Cx32 expression with EMT induction and chemosensitivity to oxaliplatin in HCC cells [59]. Compared with the HCC parental cells, the oxaliplatin-resistant cell lines presented an EMT phenotype (elongated spindle shape, loss of cell polarity, increased formation of pseudopodia), with the expression of E-cadherin (epithelial marker) being downregulated and Vimentin (mesenchymal marker) being upregulated. Moreover, oxaliplatin-resistant cells presented with significantly higher invasive and migratory activity compared to parental HCC cells, while also downregulating Cx32 expression. Aiming to define the role of Cx32 expression and its correlation with EMT and chemosensitivity, Cx32 expression was silenced via siRNA in Huh7 cell lines, resulting in EMT induction with a decrease of E-cadherin expression and an increase of Vimentin and Snail expression, while also reducing the inhibitory effect of oxaliplatin in Huh7 cells. The opposite results in EMT phenotype and markers were observed by upregulating Cx32 expression in Huh7 cells. These aforementioned results suggest that parental HCC cells with downregulated Cx32 expression present phenotypic characteristics of EMT and EMT-induced chemoresistance to oxaliplatin [59].

Moreover, Yu et al., utilizing 40 HCC tissue samples and HepG2 cell lines, developing doxorubicin-resistant HepG2/DOX cell lines and using parental HepG2 cell lines as controls, investigated the correlations of Cx32 with EMT and doxorubicin resistance [67]. Doxorubicin-resistant cells demonstrated a lower expression of the epithelial marker E-cadherin and a higher expression of the mesenchymal marker Vimentin, also showing a higher invasive and migrative activity. Cx32 overexpression in HepG2 cells as compared with HpeG2/DOX cells was observed, suggesting that Cx32 may play a role in acquired drug resistance. Aiming to establish a potential effect of Cx32 on doxorubicin-induced EMT, Cx32 expression was silenced using a siRNA and upregulated using cDNA-Cx32 in HepG2/DOX cells. A decreased expression of epithelial marker E-cadherin and an increased expression of the mesenchymal marker Vimentin were observed in the HepG2/DOX cells with downregulated Cx32, while the opposite results were observed in the HepG2/DOX cells with upregulated Cx32 and concomitant decreased invasive and migratory ability, indicating that Cx32 may be responsible for the EMT induction in doxorubicin-resistant HCC cells. 

Data obtained from the aforementioned studies regarding Cx32 association with EMT signaling network and HCC response to chemotherapeutic treatment are illustrated in Figure 2. 

#### 3.1.4. Connexin 32: Implications in Cell Survival-Proliferation Pathways and Other Tumorigenic Mechanisms

In addition to the EMT’s role in resisting apoptosis and its therapeutic exploration linked to Cx32 expression, several other mechanisms disrupt the balance between cell proliferation and death, contributing to tumorigenesis in various neoplastic processes, including hepatocarcinogenesis [94]. The JAK/STAT, PI3K/AKT, Src/FAK, and RAS/ERKs pathways are signaling pathways frequently involved in HCC resistance to apoptotic stimuli, and their expression has been shown to be upregulated in HCC tumorigenesis [95,96,97]. These pathways are currently being investigated for novel ancillary molecular markers associated with potential prognostic utility. A characteristic example is the evaluation of insulin receptor substrate 4 (IRS-4), which is considered to regulate the PI3K/Akt cascade activating several oncogenes (e.g., FER), which are characterized by tyrosine kinase activity without regulation via extracellular ligands [98]. Yet, not all of these pathways have been studied in correlation with connexins expression, but a disrupted balance between cell proliferation and cell death in correlation with gap junction protein expression in HCC has currently been investigated for Cx32. These mechanisms have not currently been extensively correlated with resistance to chemotherapy (only the regulation of the Src/FAK pathway, the association of which with Cx32 expression and chemoresistance to doxorubicin has barely been investigated), but bibliographic evidence has recently emerged [68,69,70,71,72].

Yu et al. investigated Cx32 dysregulation-induced chemoresistance to doxorubicin not only in association with the EMT signaling network but with the Src/FAK signaling pathway as well [68]. The Src/FAK pathway is continuously activated in various neoplasms [99,100,101]. Src and FAK, activating each other, create a complex which interacts with a variety of substrate proteins (e.g., CAS, paxillin, p190RhoGAP), being involved in cell proliferation processes [68]. Recent studies have suggested that Src/FAK pathway has been implicated with chemoresistance [102,103]. Based on this evidence, Yu et al. aimed to prove that Src is an important downstream regulator of Cx32 in tumor cells and is associated with chemoresistance to doxorubicin, using HCC tissue samples and 54 and HepG2 cell lines, creating HepG2/DOX cell lines resistant to doxorubicin chemotherapy [68]. After proving that the activity of the Src/FAK signaling pathway in HCC cells is regulated by Cx32 expression, Yu et al. showed that it was inhibited in HepG2-resistant cell lines, increasing their sensitivity. Considering that PI3K/Akt downregulates Src/FAK activity, after silencing Cx32 in HepG2 cells, the expression of p-AKT was found to be significantly increased, suggesting that the activity of the PI3K/Akt signaling pathway increased and regulated by Cx32 expression [68,104]. It was, overall, indicated that Cx32 expression enhances the toxicity of doxorubicin by inducing apoptosis via Src/FAK pathway downregulation [68].

The PI3K/Akt activity is dysregulated in HCC, while PTEN gene product expression levels are reduced or absent in HCC tissue samples [105,106]. Its expression, in correlation with Cx32 expression, was widely investigated in two recent studies by Zhao et al., interestingly, combined with p53 tumor suppressor gene expression, which is also disrupted in many neoplasms, including HCC [69,70,96,107,108]. In their first study, Zhao et al. used tissue samples, cell lines (HepG2, QGY-7701, SMMC-7721, and MHCC97-H), and mouse models (inoculated with MHCC97H-shCtlr and MHCC97H-shCx32 cell lines) [69]. First, they suggested that Cx32 could enhance p53 acetylation which might contribute to the upregulation of p53 transcriptional activity, resulting in the upregulation of the downstream protein CD82 [109,110,111]. Then, SMMC-7721 cells were treated with cycloheximide (CHX), an inhibitor of protein synthesis, to investigate the association between Cx32 expression levels and p53 stability, as well as the possible Cx32 induction of p53 acetylation and transcriptional activity by affecting the p53-histone deacetylase 1 (HDAC1) interaction [112,113]. The results demonstrated that Cx32 significantly prolongs the half-life of p53, and that Cx32 downregulated the level of HDAC1 protein in a dose-dependent manner in SMMC-7721 cells [69,114,115]. In a second study, Zhao et al., performing an EdU assay on HepG2 and SMMC-7721 with different levels of Cx32 expression, suggested that the expression of the proliferating cell nuclear antigen (PCNA), as well as the expression of p21 gene (which is a p53 target gene), were negatively correlated with Cx32 expression levels [69,116,117]. Moreover, the Akt signaling pathway and cyclin D1, which are considered as basic regulators of cell survival and proliferation, and specifically the levels of phosphorylated Akt (p-Akt) and cyclin D1 (p-cyclin D1) [118,119], were negatively correlated with Cx32 expression. Therefore, it is indicated that Cx32 has the ability to suppress HCC cell proliferation, via inhibiting and activity of Akt, as well as blocking the expression of cyclin D1, which is considered to be a major cell cycle regulatory protein [69].

Towards the same direction, Li et al. investigated the tumorigenic activity of PI3K/Akt signaling in correlation with Cx32 expression, regarding the potential of expanding liver CSCs, using 85 HCC tissue samples, cell lines (HCCLM3 and HepG2), and mouse models (BALB/c) inoculated or not with HCCLM3 (HCCLM3 overexpression—OE HCCLM3 empty vector—EV) [70]. After proving the higher overall survival (OS) in HCC patients with a high Cx32 expression, they studied sphere formation in HepG2 (that normally present a high Cx32 expression level) and HCCLM3 (that normally present a low Cx32 expression level) cell lines, indicating that Cx32 regulated the expansion of liver CSCs. Furthermore, when silencing Cx32 expression in HepG2 cells, mRNA expression of stemness-associated genes EpCAM, CD133, Sox9, Nanog, Oct4, and c-Myc, as well as the numbers of spheres were significantly increased, and when increasing Cx32 expression in HCCLM3 cells, the expression of the same markers and number of spheres were significantly decreased. Moreover, the regulation of PI3K/Akt signaling pathway by Cx32 was investigated by measuring phosphorylated p-Akt expression levels in HCC tissues and corresponding paracancerous tissues, and Cx32 was suggested as a regulator of the PI3K/Akt signaling activity in HCC cells [120,121]. This study proved that Cx32 regulates the expansion of liver CSCs by the PI3K/Akt signaling pathway [70]. 

The aforementioned findings are briefly illustrated in Figure 3.

#### 3.1.5. Exploring Connexin 32’s Role in Modulating Apoptosis and Necroptosis Pathways in HCC

Generally, apoptosis represents a physiological way to eliminate excess cells during both liver development and regeneration [122,123]. When apoptotic or anti-apoptotic mechanisms are dysregulated, the development and progression of tumors of the liver and the biliary tree are enhanced as a result [124,125]. Xiang et al., using HCC tissue samples and cell lines (HepG2 and SMMC-7721), aimed to investigate whether Cx32 expression is associated with anti-apoptotic mechanisms involved in HCC [71]. After proving the cytoplasmic overexpression of Cx32 in HCC tissues and its positive correlation with poor prognosis, as it was observed by other studies described herein, they silenced Cx32 expression in HepG2 cell lines (which normally present high Cx32 expression levels) and increased Cx32 expression in SMMC-7721 cell lines (which normally present low Cx32 expression levels), observing a reduced expression of the apoptosis inhibitor when Cx32 expression was increased in the first case and an increased Bcl-2 expression in the second case [126]. Aiming to further investigate the mechanisms by which Cx32 regulates apoptosis in HCC cells, they next induced apoptosis in HepG2 cells by shikonin (SHN), an apoptosis inducer. However, apoptosis was shown to be reversed when HepG2 cells were exposed to the gap junction formation inhibitor 2-APB, suggesting that under conditions of suppressed gap junction function, Cx32 expression antagonizes SHN-induced apoptosis, while in HepG2 cells cultured in low-density conditions, Cx32 silencing may exacerbate SHN-induced apoptosis [127].

After indicating that Cx32 protects HCC cells from apoptosis in a GJ-independent manner, Xiang et al. investigated potential mechanisms explaining this statement in the same study [71]. The epidermal growth factor receptor (EGFR) signaling pathway is considered to play a pivotal role in cell proliferation, survival, and apoptosis, being involved in many types of neoplasms and even being associated with sensitivity to chemotherapeutic evaluation [128,129,130]. The EGFR signaling pathway was chosen by Xiang et al. as a potential network involved in HCC cells protection from apoptosis by Cx32. Suggesting that Cx32 expression was low in HCC specimens, a significant increase of EGFR, p-EGFR, p-STAT3, and p-Erk1/2 was observed when increasing Cx32 expression, and a significant decrease of EGFR, p-EGFR, p-STAT3, and p-Erk1/2 was observed when downregulating Cx32 expression in HCC cell lines. It is noteworthy that the expression levels of total STAT3 and Erk1/2 remained the same in these two cases. These results suggest that the expression levels of Cx32 and EGFR were positively correlated in HCC specimens and cell lines, and overexpressed Cx32 is involved in signaling pathway activation. [71]. Src kinase is considered to enhance tumorigenic activity after interacting with EGFR. Upon conducting experiments of silencing and overexpressing Cx32 in cell lines and confirming the results using mouse models, Cx32 upregulated and activated EGFR by enhancing the expression of Src [71].

Necroptosis is an alternative mode of regulated cell death, mimicking features of apoptosis and necrosis [131]. Necroptosis is triggered by death receptors (e.g., Fas/FasL), Toll-like receptors (TLR4 and TLR3), and cytosolic nucleic acid sensors (e.g., RIG-I, STING), which induce type I interferon (IFN-I) and TNFα production [131]. The majority of these mechanisms trigger NFκB-dependent proinflammatory signals in favor of survival [131,132]. RIPK3 (previously well recognized as regulator of inflammation, cell survival, and disease) and its substrate MLKL, are the main regulators of this pathway [133,134,135]. Necroptosis pathway dysregulation is considered to be associated with a variety of pathological conditions, including neoplasms, neurodegenerative diseases, as well as inflammatory diseases [133,136]. Xiang et al. conducted another analysis in 30 human HCC specimens, this time investigating the way Cx32 expression affects necroptotic signaling network and necroptosis biomarkers (RIPK1, phosphorylated RIPK1–p-RIPK1, and phosphorylated MLKL–p-MLKL) expression levels [72]. Using Western blotting, they proved that Cx32 expression was significantly positively correlated with the expression of p-RIPK1, RIPK1, and p-MLKL. After HCC specimens’ exposure to shikonin (SHN), which induces necroptosis, it was shown that overexpressed Cx32 increased the SHN-induced interaction of RIPK1-RIPK3 and membrane translocation of MLKL, resulting in the formation of necrosome and membrane damage, while Cx32 downregulation alleviated necroptosis [127,137,138,139]. Considering the role of caspase 8 in regulating apoptosis by activating downstream caspase cascades and in inhibiting necroptosis by cleaving RIPK1, Xiang et al. proved that Cx32 activates necroptosis in HCC cells via mediating the inactivation of caspase 8 [72,140,141]. 

Evidence observed from the aforementioned studies regarding the association of Cx32 with apoptotic signaling pathways of HCC, as well as the apoptosis and proliferation pathways by which Cx32 suppresses metastasis and proliferation of HCC cells, are illustrated in Figure 3. 

#### 3.1.6. Connexin 32 and Pre-Cancerous HCC-Related Conditions 

Cx32 has been barely studied in terms of its association with pre-cancerous states and factors accelerating hepatocarcinogenesis, with only two studies being currently available in literature [73,74].

Alcohol has been classified as a group 1 carcinogen by the International Agency for Research on Cancer, inducing HCC and other types of human neoplasms [142,143]. Towards this direction, Kato et al. aimed to correlate ethanol consumption with Cx32 expression in HCC [73]. In vivo, ethanol treatment has been shown to enhance chemically induced hepatocarcinogenesis using animal models; however, other studies dispute this statement, and the detailed molecular mechanisms by which ethanol contributes to hepatocarcinogenesis has not been established yet [144,145,146]. Kato et al., using mouse models, and specifically Cx32 dominant negative transgenic (Tg) and wild-type (Wt) mouse models which were given 1 or 5% ethanol or water ad libitum for 16 weeks after an intraperitoneal injection of 200 mg/kg diethyl nitrosamine, suggested that dysregulated Cx32 expression may promote ethanol-related hepatocarcinogenesis, and that Cx32 dysfunction compared with exposure to ethanol decreases Dusp1 expression leading to Erk activation in glutathione S-transferase placental form (GST-P)-positive foci, enhancing tumorigenic activity [73]. 

Metabolic dysfunction-associated steatotic liver disease (MASLD), formerly known as non-alcoholic fatty liver disease (NAFLD), is a chronic liver disease with a wide spectrum of activity, from simple steatosis to steatohepatitis [7,8,147]. Sagawa et al., using mouse models, specifically Cx32 dominant negative transgenic (Cx32ΔTg) and wild-type (Wt) mouse models, which were given diethylnitrosamine and fed methionine-choline-deficient diet (MCDD) or MCDD with luteolin for 12 weeks, as well as liver tissue samples for the histologic analysis of non-alcoholic steatohepatitis (NASH), aimed to define the role of Cx32 and the chemopreventive effect of luteolin, an antioxidant flavonoid, on the progression of NASH and NASH-related hepatocarcinogenesis [74]. Results indicated higher steatosis with more severe fibrosis on Cx32ΔTg rats and decreased expression of Cx32 in steatohepatitis on Wt rats, suggesting that Cx32 may be a protective factor against NASH progression. Moreover, luteolin seems to have the ability to prevent NASH progression and NASH-related hepatocarcinogenesis, demonstrating tumor-protective properties. These findings suggest that Cx32 is a potential therapeutic target and that luteolin represents a promising chemopreventive agent for NASH and NASH-related hepatocarcinogenesis [74]. 

Detailed insights into the role of Cx32 expression in HCC, along with the potential regulatory mechanisms and therapeutic implications, are summarized in Table 1.

### 3.2. Connexin 43 (Cx43)

Cx43 expression forms the basis of GJs in normal liver cells and is also expressed in HCC cells, presenting high expression levels and differentiating it from Cx32, with a single study, however, suggesting a lower Cx43 expression in HCC tissue samples [62,63]. This discrepancy is explained later herein. Cx43 is mainly located in the cytoplasm and barely expressed in areas connecting non-adjacent cells on the cell membrane [148]. In HCC, studies have shown that Cx43 expression is associated with the presence of histological differentiation, multiple foci, vascular tumor thrombosis, and early recurrence [149]. However, it was reported in a single study on human HCC tissue that Cx43 protein was not detectable in normal liver, but it could be found in HCC cells, suggesting that certain tumors or transformed cells have normal levels of GJs and that the lack of functional GJs is not a general feature of malignancy [150]. Cx43′s clinical significance in HCC has been investigated in a limited number of studies [63,149,151,152]. 

The first study focusing on Cx43′s expression levels and characteristics and on their involvement on HCC metastatic capacity was conducted by Wang et al., who used 38 HBV-HCC tissue samples investigating Cx43 expression in correlation with vascular epithelial growth factor (VEGF), a-fetoprotein (AFP) levels, prognosis, and metastatic potential [149]. Cx43’s s expression mainly occurred in HCC cytoplasm (49.1%), and this expression was lower in HCC compared with para-cancerous tissue (72.8%) and cirrhosis tissue (92.4%). A negative correlation between Cx43 and VEGF expression levels was observed, indicating a poor prognosis in patients with low Cx43 levels, like HCC patients. Regarding AFP expression levels, they are also negatively correlated with Cx43 expression, with low-AFP samples highly (47.1%) expressing Cx43, in comparison with high-AFP samples (28% of which expressed Cx43). Interestingly, in HBV-HCC specimens with a low AFP level, the positive Cx43 expression was mostly obtained in specimens of patients with distant metastases, as compared with those with liver-only metastases or no metastases. These results reveal that Cx43 expression in HBV-HCC is a potential marker of advanced disease, highlighting its significance as a novel prognostic marker [149].

Another study, conducted by Ogawa et al. using HCC cell lines (HSU-C1, -C5F, -C6, -N1, and -L2) and mouse models (inoculated with HSU- C1, -C5F, -C6, -N1, and -L2 cells), investigated the possible regulation of invasion and metastasis in HCC by Cx43 expression [151]. From the cell lines used, -N1 and -L2 were proved to have metastatic potential to the lung. Cx43 expression in cell lines used was higher than in normal liver tissue and especially in -N1 and -L2 cell lines. Cx43 expression in -C1, -C5F and -C6 cell lines was not associated with lung metastases, while in -N1, the metastatic potential to the lung was found to be same as in normal liver tissue, and in -L2, it was found to be significantly higher. Silencing Cx43 expression in -L2 cell lines with a siRNA did not affect proliferation and apoptosis; however, it decreased invasive and migrative capacity. Conducting the same experiments in vivo in mouse models inoculated with the same cell lines, confirmed these findings. Evidence obtained from this study suggests that Cx43 partially regulates tumor invasion and metastasis. Moreover, molecular targeting of Cx43 could suppress metastases in Cx43 overexpressing tumors.

Finally, a study conducted by Wang et al., using HCC specimens and SMMC-7721 cell lines, aimed to identify the downstream target genes of Cx43 by Human Transcriptome Array [152]. Therefore, a Cx43 overexpression plasmid to overexpress Cx343 in SMMC-7221 cells was constructed, and a gene microarray was generated consisting of Cx43-overexpressed HCC cells, transfected with the constructed plasmid and negative controls. Bioinformatic analysis revealed that *RALA* and *SRC* genes were highly expressed in HCC tissues. Cx43 silencing with a siRNA in SMMC-7221 cell lines resulted in an upregulation of *RALA* and *SRC* genes, which transduce signals involved in the control of a variety of cellular processes, including proliferation, differentiation, motility, and adhesion, rendering them novel therapeutic targets in HCC with high Cx43 expression. However, further in vivo research is needed to confirm these findings before translating them into clinical practice [152].

Detailed insights into the role of Cx43 expression in HCC, along with the potential regulatory mechanisms and clinical implications, are summarized in Table 2.

### 3.3. Connexin 26 (Cx26)

Cx26 expression is not only observed in normal liver tissue, but it can be related with neoplastic procedures due to mechanisms of tumor cells differentiation, especially EMT of human highly invasive tumor cells [22,23,63]. Cx26 expression in HCC has been studied in comparison to normal controls, being lower in HCC compared to normal cells at mRNA level [63]. Regarding potential Cx26 expression-related factors, mRNA levels were significantly correlated with cell differentiation, but not with gender, age, serum AFP level, chronic HBV or HCV carriage, tumor size, coexisting cirrhosis, encapsulation, vascular permeation, daughter nodules, tumor necrosis, or tumor hemorrhage [63]. Most importantly, Cx26 has been studied in terms of its association with sensitivity to current HCC treatment options, providing data on its prognostic and therapeutic potential [22,23].

To our knowledge, Cx26 is the only of the HCC-related connexins that has been related with sensitivity to radiotherapy [22]. Currently, radiotherapeutic options in HCC include three-dimensional conformal radiotherapy (3D-CRT), stereotactic body radiotherapy (SBRT), hypo-fractionated radiotherapy (HFRT), and proton and heavy ion radiotherapy, none of which have shown serious effects in improving long-term survival in HCC, which highlights the need of improving our knowledge on its therapeutic effect [153]. A recent study by Li et al., using human HCC tissue and cell lines with different expression level of Cx26 (pG2 with low Cx26 expression and SK-hep-1 with high Cx26 expression), investigated the Cx26 correlation with survival, focusing on the radiosensitivity of HCC cells in correlation with Cx26 level expression and proposing the activation of the MAPK and NF-κΒ signaling pathway as a potential mechanism associated with the observed difference in radiosensitivity. Specifically, tissue sample analysis showed that Cx26 expression is positively associated with survival. Cx26 presented no significant difference in expression between irradiated and control cells, while low Cx26 expression results in lower radiosensitivity. The MAPK signaling pathway, activated by growth factors and tumor promoters, is essential in controlling cell survival and cell death in many types of human neoplasms and is considered to play an important role in protein intracellular channels, including gap junctional communication [154,155]. Li et al. proved different activation of MAPK and NF-κB in the different type of cell lines (with radiosensitive cell lines presenting lower expression), which is strengthened in cases of Cx26-upregulated expression for HepG2 cell lines.

Cx26 effects on chemosensitivity to oxaliplatin have been investigated in vitro. Yang et al. used normal (LO2) and HCC (SMMC-7721) cell lines to perform expression analysis in three connexin members (Cx26, Cx32, Cx43). Specifically, the inhibition of Cx26, decreased oxaliplatin cytotoxicity. Additionally, the upregulation of Cx26 expression enhanced GJ formation and ultimately oxaliplatin toxicity. They found that oxaliplatin cytotoxicity in high-density cultures rich in GJ formation was higher than that of low-density cultures without GJ formation, showing that cell density affected oxaliplatin cytotoxicity. Endothelial growth factor (EGF) and increased adherence proteins were proposed as possible molecules affecting chemotherapy effects in areas with GJ formation [23].

Detailed insights into the role of Cx26 expression in HCC, along with the potential regulatory mechanisms and therapeutic implications, are summarized in Table 3.

## 4. Discussion

In this review, we showcase that HCC is a type of cancer characterized by complexity regarding connexin expression [156,157,158]. As a general rule, Cx32 and Cx26 expression at a protein and mRNA level is significantly decreased in HCC tissues and cell lines, compared to normal liver tissue or cell lines. An exception to this general expression pattern is Cx43, which increases in HCC at an mRNA level, except in one study which demonstrated a lower expression of Cx43 in HCC [62,149,151,152]. This discrepancy in Cx43 expression could be partially explained by the large amount of connective tissue in HCC specimens, by the various liver pathologies of HCC patients under study, and due to the appearance of isozymes (e.g., aldolase and γ-glutamyl transpeptidase) in human HCC, which have been described as the potential culprit for Cx43 expression level increase [149]. There are various potential mechanisms which may be involved in changes of connexin expression, including the rapid proliferation of tumor cells, changes in the interactions between host and tumors, aberrant localization of connexins, and changes in connexin expression during tumor differentiation. An insufficient connexin quantity in a liver specimen might be a factor predicting HCC development [75,159].

A significant number of molecular mechanisms involved in HCC development and progression are compromising the balance between survival and apoptotic signals in the pre-neoplastic hepatocytes [160]. This is partially a result of dysregulated expression of pro-apoptotic molecules, but the balance between death and survival is mainly disrupted due to overactivation of the anti-apoptotic signaling pathways. Various observations were made considering Cx32, which mainly presents antitumorigenic effects [161]. In more detail, Cx32 downregulates the PI3K/Akt pathway, inhibits cell migration and invasion via p53 regulation, and enhances necroptosis via RIPK3/MLKL regulation [68,69,72]. Interestingly, when Cx32 was studied in terms of its non-junctional role, it was revealed that it has the potential to prevent apoptosis by promoting tumor survival via the EGFR signaling pathway [71]. The role of Cx32 in HCC cell survival and death seems to be complex and to depend on the corresponding activated signaling pathways; even in the same type of neoplasm, Cx32 displays either apoptotic or anti-apoptotic effects regarding its interactions. Overall, it is important to highlight the crucial Cx32 role in suppressing the progression of human HCC by inhibiting cell proliferation and invasiveness. These data suggest that modulation of Cx32 could represent a future therapeutic strategy for the treatment of HCC [68,69,72].

HCC is characterized by poor outcomes, mainly due to the high resistance of HCC to the available chemotherapeutic options [162]. Chemoresistance is associated with more than 100 involved genes, as well as a variety of synergistic mechanisms. HCC is a highly vascularized tumor, which creates hypoxic conditions, associated with increased invasiveness and poorer prognosis [163]. Moreover, the activity of signaling pathways, for example, the aberrant Wnt/β-catenin pathway and the increased activity of the PI3K/Akt/mTOR signaling pathway, is associated with poor prognosis, early recurrence, and reduced survival in HCC [164,165]. It is worth mentioning that these signaling pathways were described herein in correlation with connexins expression [70]. Exosomes, EMT, ATP binding box transporters, and apoptotic mechanisms are also involved in decreased sensitivity to current, and even targeted, therapeutic options [166,167]. It seems that the chemoresistance of HCC cells is partially explained by many mechanisms at a molecular and genetic level. Thus, the application of sensitizing tools and investigating approaches that are correlated with the aforementioned mechanisms may be a potential promising line of action to overcome HCC resistance to therapy.

In addition, the direct correlation of connexins with the response to available treatment options has been recently investigated. Specifically, Cx32 expression has been associated with chemosensitivity to oxaliplatin and doxorubicin chemotherapy, and Cx26 has been associated with chemosensitivity to oxaliplatin chemotherapy and radiosensitivity. Generally, it has been demonstrated that injury and death signals induced by platinum-based agents are transferred between adjacent tumor cells via GJs, expanding their cytotoxic activity and suggesting that the final anti-tumor effect of chemotherapeutic drugs depends largely on the amount of tumor gap junctions [168,169]. However, as previously described, connexin expression in HCC is low, limiting the transmission of chemotherapy toxic signals between adjacent cells and resulting in reduced chemosensitivity [62,63]. This explains the significantly positive correlation between Cx32 and Cx26 expression and chemosensitivity to oxaliplatin and doxorubicin, as it has been reported in recent studies [23,59,66,68,70]. It is also important that the majority of these studies focused on EMT, which has been proposed as a potential target to overcome cancer drug resistance, especially in HCC tumors with low connexin levels [23,59,66,68,70,170].

A schematic presentation of the most important connexin expression correlations with HCC-related factors presented is this review, is given in Figure 4.

## 5. Conclusions—Future Perspectives

The study of the pathogenetic mechanisms of tumorigenesis, invasion, metastasis, and patient response to available therapeutic strategies is an ever-evolving field of research, aiming to prolong overall survival rate and improve patients’ quality of life. The detection of new molecules involved in the processes of hepatocarcinogenesis may bring into spotlight novel diagnostic and prognostic biomarkers and new promising therapeutic targets. Towards this direction, identifying the expression patterns of connexins in HCC and investigating their correlation with tumor characteristics, signaling pathways, and neoplastic processes, as well as establishing their potential role as biomarkers that predict responses to chemotherapy and/or radiotherapy, may contribute to harnessing further molecular targeted therapies in the context of personalized medicine in HCC.

## Figures and Tables

**Figure 1 cancers-16-01533-f001:**
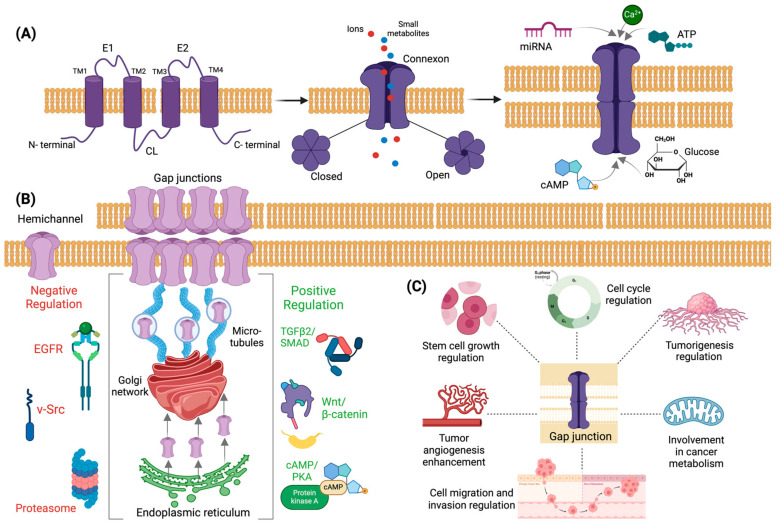
Connexins’ structure, positive and negative regulatory mechanisms, and their involvement in tumorigenesis. (**A**) Connexins consist of four hydrophobic transmembrane domains (TM1–TM4), one intracellular (cytoplasmic) loop (CL) and two extracellular loops (E1–E2), and N- and C-terminal cytoplasmic domains. They permit the exchange of small molecules, including ions (K^+^, Ca^2+^) second messengers (cAMP, cGMP, inositol triphosphate (IP_3_)), small metabolites (glucose), antigens, and microRNAs. (**B**) Connexins are synthesized in the endoplasmic reticulum and are transported to the Golgi apparatus and from there, via microtubules, to the cell membrane, either in cell-to-cell contact areas, forming gap junctions, or to unopposed areas of the cell membrane. There are various mechanisms regulating connexins’ expression and activity from their synthesis to their transportation to the cell membrane, positively (e.g., TGFβ2/SMAD, Wnt/β-catenin, cAMP/PKA) or negatively (e.g., proteasome, EGFR, v-Src) (**C**) Connexins are considered to be regulators of cell cycle, tumorigenesis, cancer metabolism, cell migration and invasion, tumor angiogenesis, and stem cell growth. Created with BioRender.com.

**Figure 2 cancers-16-01533-f002:**
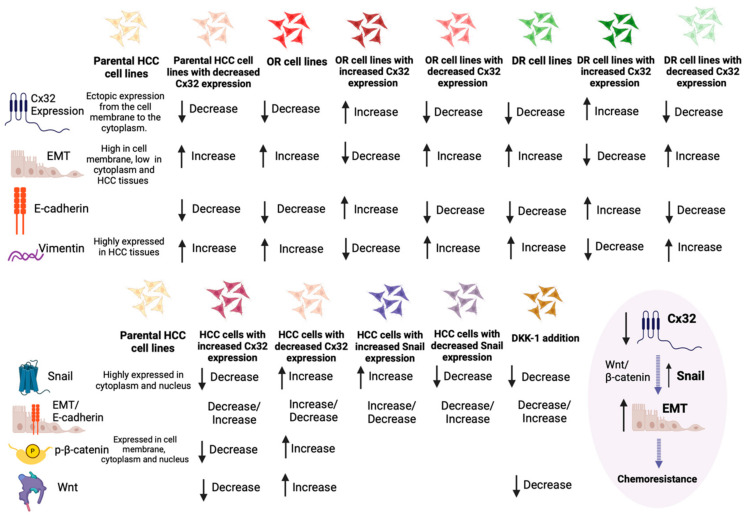
Schematic presentation of current evidence on Cx32 association with EMT and the HCC chemotherapeutic resistance with oxaliplatin or doxorubicin. Cx32 expression is negatively correlated with chemoresistance and EMT phenotype enhancement, while the downregulation of Cx32 upregulates Snail expression and promotes EMT via the activation of the Wnt/β-catenin signaling pathway. Thus, the Cx32/β-catenin/Snail pathway is as a potential therapeutic target in advanced HCC. (HCC: hepatocellular carcinoma; OR cells: oxaliplatin-resistant cells; DR cells: doxorubicin-resistant cells). Created with BioRender.com.

**Figure 3 cancers-16-01533-f003:**
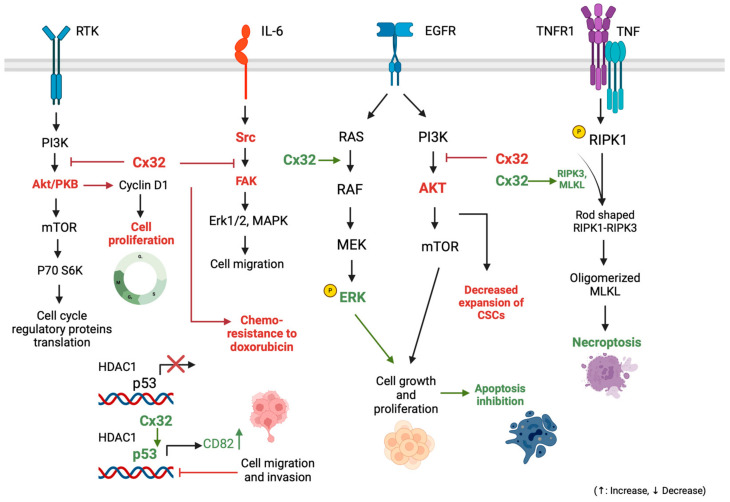
Schematic presentation of current evidence on Cx32 association with signaling pathways regulating proliferative and cell death processes. Cx32 inhibits the PI3K/Akt pathway, sensitizing HCC cells to doxorubicin, inhibits cell migration and invasion via p53 regulation, and enhances necroptosis via RIPK3/MLKL regulation. Regarding cell growth and proliferation, Cx32, despite activating ERK via the EGFR pathway, decreases CSCs expansion via the PI3K/Akt pathway blockage, demonstrating an anti-tumorigenic effect. Created with BioRender.com.

**Figure 4 cancers-16-01533-f004:**
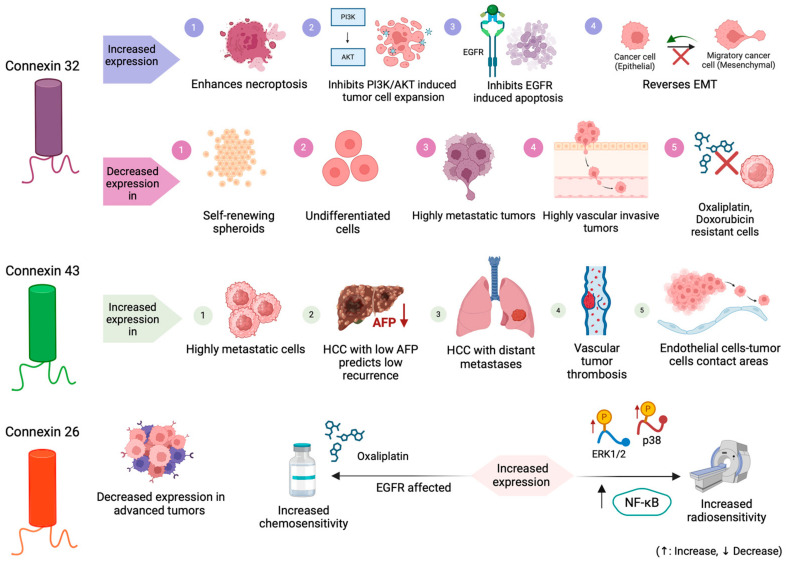
A schematic overview of connexin expression’s correlations with HCC-related factors. Created with BioRender.com.

**Table 1 cancers-16-01533-t001:** The role of Cx32 in HCC in terms of expression, regulation of expression and clinical/therapeutic implications, along with the proposed regulatory mechanisms.

Connexin Studied and Reference	Material Studied	Expression—Regulation of Expression	Clinical Implication	Mechanisms Involved
Ma et al. (2002) [62]	Cell lines (HCC cell lines: HHCC, SMMC-7721 and normal liver cell line: QZG) [62]	1. Low expression in HCC samples compared to normal liver samples [62] 2. Downregulation of expression during tumorigenesis because aberrant localization to the cytoplasm [62]	Not applicable	Not applicable
Sheen et al. (2004) [63]	Tissue samples (25 HCC samples and 15 normal controls) [63]	1. Lower expression in HCC compared to normal cells [63] 2. Expression significantly correlated with cell differentiation [63] 3. No correlations between expression and gender, age, serum AFP level, chronic HBV or HCV carriage, tumor size, coexisting cirrhosis, encapsulation, vascular permeation, daughter nodules, tumor necrosis, or tumor hemorrhage [63]	Upregulated expression is associated with high recurrence end recurrence-related mortality [63]	Not applicable
Li et al. (2007) [64]	HuH7 cell lines with overexpressed Cx32 by doxycycline withdrawal (Tet-off HuH7 cells) and Li-7 HCC cell lines [64]	1. Expression occurs mainly in the cytoplasm, but not in cell-to-cell contact areas, suggesting intracellular Cx32 sorting to the plasma membrane [64] 2. Overexpression of cytoplasmic Cx32 protein enhances proliferation, motility, and invasiveness in a gap junction independent manner [64]	Not applicable	Not applicable
Kawasaki et al. (2011) [65]	HuH7 HCC cell lines with overexpressed Cx32 by doxycycline withdrawal (Tet-off HuH7 cells) and HuH7 Tet-off mock cells [65]	1. Higher Cx32 expression is associated with larger SP fractions [65] 2. Cytoplasmic accumulation of Cx32 expands CSC population development, elevating CSC renewal rate and enhancing spheres formation [65]	Not applicable	Not applicable
Yang et al. (2017) [66]	1. Tissue samples (76 HCC samples and 20 normal controls) 2. Cell lines (normal hepatic cell line: LO2, HCC cell lines: HepG2, Huh7 and SMMC-77210) 3. Mouse models (Huh7-hCx32 cells or Huh7-vec cells inoculated) [66]	1. Lower expression in HCC tissue compared to normal tissue [66] 2. Ectopic expression from the cell membrane (where is normally expressed) to the cytoplasm in HCC tissue [66] 3. Expression negatively correlated with histological grade and lymph node metastasis [66] 4. Expression not correlated with age, sex, tumor size, TNM stage, liver disease medical history, vascular embolus [66]	1. Downregulation of expression is associated with a metastatic phenotype [66] 2. Downregulation associated with chemoresistance to oxaliplatin [66] 3. Targeting Cx32 proposed as a potential target to overcome oxaliplatin resistance [66]	1. Downregulation of expression enhances cell migration and invasion [66] 2. Downregulation enhances EMT [66] 3. EMT regulated by Cx32 in HCC cells is mediated by Snail signaling pathway [66]
Yang et al. (2019) [59]	HepG2, Huh7, and SMMC-7721 cell lines (oxaliplatin-resistant cell lines development and parental HCC lines as controls) [59]	1. High expression in the cell membrane in adjacent non-tumor tissues [59] 2. Low expression in the HCC cells, mainly expressed in the cytoplasm [59] 3. Low expression in oxaliplatin resistance cells [59]	1. Downregulation associated with chemoresistance to oxaliplatin [59] 2. Targeting Cx32 proposed as a potential target to overcome oxaliplatin resistance [59] 4. Expression positively correlated to chemosensitivity to oxaliplatin [59]	1. EMT phenotype and low Cx32 expression in oxaliplatin-resistant cells [59] 2. Downregulating Cx32 expression resulted in EMT induction [59] 3. Cx32 expression positively correlated with E-cadherin and negatively correlated with the expression of Vimentin and Snail [59]
Yu et al. (2017-1) [67]	40 HCC tissue samples, HepG2 cell lines (oxaliplatin-resistant HepG2/DOX cell lines and parental HepG2 cell lines as controls) [67]	1. Lower expression in HCC tissue compared to para-cancerous tissue [67] 2. Expression positively correlated with E-cadherin expression (in HCC tissues compared to para-cancerous tissue) and negatively correlated with Vimentin expression (higher in HCC tissue) [67] 3. Low expression in doxorubicin-resistant cells [67] 4. Expression may play a role in acquired drug resistance [67]	1. Downregulation associated with chemoresistance to doxorubicin [67] 2. Targeting Cx32 proposed as a potential target to overcome doxorubicin resistance [67]	1. Downregulating Cx32 expression resulted in EMT induction [67] 2. Cx32 expression was positively correlated with the expression of E-cadherin and negatively correlated with the expression of Vimentin [67]
Yu et al. (2017-2) [68]	54 HCC tissues, HepG2 cell lines [68]	1. Lower expression in HCC compared to para-cancerous normal liver tissues [68] 2. Expression positively correlated with differentiation degree [68] 3. Low expression in doxorubicin-resistant cells [68]	Downregulation associated with chemoresistance to doxorubicin Targeting Cx32 and Src/FAK signaling pathway is proposed as a potential target to overcome doxorubicin resistance [68]	Cx32 affects the chemoresistance to doxorubicin via the regulation of the activity of Src/FAK signaling pathway. Expression positively correlated to chemosensitivity to doxorubicin [68]
Zhao et al. (2015) [69]	1.24 HCC tissue samples and 24 normal liver tissue samples [69] 2. Cell lines (HepG2, QGY-7701, SMMC-7721, MHCC97-H) [69] 3. Mouse models (inoculated with MHCC97H-shCtlr and MHCC97H-shCx32 cell lines) [69]	Lower expression in HCC tissue compared to normal tissue [69]	Downregulation associated with a poor prognosis [69]	1. Cx32 expression suppressed invasion and migration via p53 pathway [69] 2. Cx32 upregulates CD82 expression via p53 [69] 3. Cx32 suppressed HCC progression in vivo [69]
Li et al. (2022) [70]	1.85 HCC tissue samples Cell lines (HCCLM3 and HepG2) [70] 2. Mouse models (BALB/c) inoculated or not with HCCLM3 (HCCLM3 over expression—OE HCCLM3 empty vector—EV) [70]	1. Expression downregulated in CSCs [70] 2. Expression regulated CSCs expansion [70]	Expression associated with a poor prognosis. Targeting Cx32 potentially inhibits the invasion and metastasis of liver cancer cells and reverses drug resistance [70]	Cx32 regulates the activity of the PI3K/Akt signaling pathway in HCC cells and CSCs expansion by the PI3K/Akt signaling pathway [70]
Xiang et al. (2019) [71]	1. 96 HCC tissue samples 2. Cell lines (HepG2 and SMMC-7721) [71]	1. Overexpression and internalization in HCC [71] 2. Expression positively correlated with Bcl-2 expression and negatively correlated with Bax and Bak expression [71]	1. Expression associated with a poor prognosis [71] 2. Cx32 presents an intrinsic anti-apoptotic effect in HCC cells, in case of impaired gap junctions’ function [71]	Interaction between Cx32 and Src contributes to the EGFR activation which mediates the Cx32 anti-apoptotic effect in HCC cells [71]
Xiang et al. (2021) [72]	1. HCC tissue samples [72] 2. PLC/PRF/5 and SMMC-7721 cell lines [72] 3. Mouse models (BALB/c-nu mice) inoculated with PLC-Vector cells and PLC-shCx32 cells (PLC-Vector+Vehicle, PLC- shCx32+Vehicle, PLC-Vector+SHN, and PLC-shCx32+SHN) [72]	1. Expression positively correlated with expression levels of necroptosis biomarkers (RIP1, p-RIP1, and p-MLKL) [72] 2. Downregulated expression suppresses SHN-induced necroptosis in PLC/PRF/5 cells [72] 3. Upregulated expression enhances SHN-induced necroptosis by upregulation of RIP1, RIP3, and MLKL in HCC cells [72]	1. Cx32 may be involved in a therapeutic strategy consisting of necroptosis inducers, which may be effective in HCC patients with high Cx32 expression levels [72] 2. The overexpression of Cx32 could be a potential therapeutic biomarker in HCC [72]	1. Cx32 enhances the c-FLIPs and downregulates FADD, resulting in caspase 8 inactivation and protection from RIP1 and RIP3 caspase 8-mediated cleavage [72] 2. Cx32 interacts with Src and contributes to the Src-mediated phosphorylation of caspase 8, resulting in suppression of caspase 8 and activation of necroptosis [72] 3. Cx32 knockdown suppresses necroptosis in vivo [72]
Kato et al. (2016) [73]	Mouse models: -Cx32 dominant negative transgenic (Tg) -wild-type (Wt), which were given 1% or 5% ethanol or water ad libitum for 16 weeks after an intraperitoneal injection of 200 mg/kg diethyl nitrosamine [73]	Downregulation of expression positively correlated with Dusp1 and Dusp4 downregulation of expression in a protein level and with Dusp1 downregulation of expression in a mRNA level, in Tg mouse models [73]	Dysregulated expression may promote ethanol-related hepato-carcinogenesis [73]	Cx32 dysfunction compared with exposure to ethanol decreases Dusp1 expression leading to Erk activation in GST-P positive foci, enhancing tumorigenic activity [73]
Sagawa et al. (2015) [74]	Mouse models: -Cx32 dominant negative transgenic (Cx32ΔTg) -wild-type (Wt), which were given diethylnitrosamine and fed methionine–choline-deficient diet (MCDD) or MCDD with luteolin for 12 weeks [74]	Expression negatively correlated with NASH development [74]	1. Dysregulated expression may promote steatohepatitis and fibrosis [74] 2. Reduced expression of Cx32 associated with NASH is prevented by luteolin [74]	Not applicable

**Table 2 cancers-16-01533-t002:** The role of Cx43 in HCC in terms of expression, regulation of expression, and clinical implications, along with the proposed regulatory mechanisms.

Connexin Studied and Reference	Material Studied	Expression	Clinical Implication	Mechanisms Involved
Ma et al. (2002) [62]	Cell lines (HCC cell lines: HHCC, SMMC-7721 and normal liver cell line: QZG) [62]	1. Low expression in HCC samples compares to normal liver samples, except expression in SMMC-7721 cells [62] 2. mRNA expression not significantly different between HCC and normal liver samples [62]	Not applicable	Not applicable
Sheen et al. (2004) [63]	Tissue samples (25 HCC samples and 15 normal controls) [63]	Higher expression in HCC tissue samples [63]	Expression not correlated with recurrence and mortality [63]	Not applicable
Wang et al. (2013) [149]	38 HBV-HCC tissue samples [149]	1. Expression positively correlated with histological differentiation, multiple foci, and vascular tumor thrombosis [149] 2. Expression negatively correlated with MVD-CD105 and VEGF expression levels, and with distant metastases [149]	1. Cx43 downregulated expression associated with poor prognosis in HBV-HCC patients with a low AFP level [149] 2. Survival rates are positively correlated with Cx43 expression [149]	Not applicable
Ogawa et al. (2012) [151]	1. HSU-C1, C5F, -C6, -N1 and -L2 cell lines [151] 2. Mouse models (inoculated with C5F, C6, N1 and L2 cells) [151]	1. Higher expression in high metastatic cell lines (N1 and L1) compared with low metastatic cell lines (C1, C5F, C6) [151] 2. Higher expression endothelial cells–tumor cells contact areas [151]	1. Silencing with siRNA resulted in decrease of lung metastasis in mouse models [151] 2. Silencing with siRNA suppressed invasion and migration of L2 cell line (accompanied by MMP-9 decrease) [151]	1. MMP-9 association with HCC metastatic capacity explains the decreased metastatic capacity as a result of MMP-9 decrease via Cx-43 silencing [151] 2. Higher expression of Cx-43 in highly metastatic HCC cells results in GJ formation between tumor cells and endothelial cells [151]
Wang et al. (2019) [152]	80 HCC tissue samples Cell lines (SMMC-7221) [152]	Expression negatively associated with RALA and SRC genes expression [152]	Targeting the downstream mechanisms regulated by Cx43 (RALA, SRC: target genes) is a potential therapeutic strategy for advanced HCC [152]	RALA and SRC are significantly upregulated in CX43-silenced HCC cells and significantly associated with HCC survival [152]

**Table 3 cancers-16-01533-t003:** The role of Cx26 in HCC in terms of expression, regulation of expression, and therapeutic implications (response to chemotherapy and radiotherapy).

Connexin Studied and Reference	Material Studied	Expression—Regulation of Expression	Mechanisms Involved	Clinical Implication
Sheen et al. (2004) [63]	HCC tissue samples (25) compared to normal controls (15) [63]	1. Lower expression in HCC compared to normal cells.Expression significantly correlated with cell differentiation [63] 2. No correlations between expression and gender, age, serum AFP level, chronic HBV or HCV carriage, tumor size, coexisting cirrhosis, encapsulation, vascular permeation, daughter nodules, tumor necrosis, or tumor hemorrhage [63]	Not applicable	Not applicable
Li et al. (2021) [22]	HCC tissue and cell lines (pG2 with low Cx26 expression and SK-hep-1 with high Cx26 expression [22]	Same expression in irradiated and control cells Expression positively associated with survival [22]	Overexpression positively correlated with overactivation of MAPK and NF-κB signaling pathways [22]	Expression positively correlated with radiosensitivity [22]
Yang et al. (2016) [23]	Cell lines (normal liver cell line: LO2 and HCC cell lines: SMMC-7721) [23]		Endothelial growth factor and increased adherent proteins affect chemosensitivity [23]	Cx26 inhibition decreases oxaliplatin cytotoxicity [23]

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
