# Peer review of "Understanding the Role of Connexins in Hepatocellular Carcinoma: Molecular and Prognostic Implications"

_cancers, 2024, doi:10.3390/cancers16081533_

Round 1
Reviewer 1 Report
Comments and Suggestions for Authors
In the manuscript presented by Papadakos et al. authors aimed to describe the molecular biochemical background and prognostic value of Connexins in hepatocellular carcinoma (HCC). In more detail, authors comprehensively reviewed on both pro- and anti-tumorigenic effects of three members of the connexin family (Cx32, Cx43 and Cx26), their role in cell migration, invasion, and proliferation, affecting metastasis and chemo- and radiation resistance of HC cells.
In summary, there are no major issues regarding this manuscript, as the issue addressed is of clinical relevance and the authors comprehensively reviewed the current knowledge. Overall, the review is well-designed, and the text is supported by informative arranged figures. There are, however, some minor issues as mentioned successively, aiming to further improve readability.
Minor points of criticism:
1. Graphical abstract is misleading in part as authors did not review in detail on connexin(s) expression in hepatocytes, Kupfer cells, Stellate cells etc. or on the impact of connexins in liver diseases.
2. Title and abstract section: Authors did not explicitly refer to diagnostic or therapeutic implications of connexin expression in HCC. Thus, the title should be more precise: e.g. “…mechanistic and prognostic implications”.
3. Tables are complex and confusing. Authors should delete the column “chemosensitivity” in table 1 and table 2 as this information are redundant to the column “clinical implications” and should delete "chemosensitivity and radiosensitivity" in table 3.
4. Moreover, organization of the tables should be improved, as it is difficult to separate the information given in the references cited.
Author Response
Dear Reviewer,
thank you for your comments, your time is highly appreciated.
Point 1: Graphical abstract is misleading in part as authors did not review in detail on connexin(s) expression in hepatocytes, Kupfer cells, Stellate cells etc. or on the impact of connexins in liver diseases.
Response 1: We updated the graphical abstract. We excluded the part referring to liver disease according to your comment, as we are not clarifying these data in our review. We updated the part of the graphical abstract presenting the expression of connexin in normal liver, to come in concordance with the reference in our manuscript, lines 79-81.
Point 2: Title and abstract section: Authors did not explicitly refer to diagnostic or therapeutic implications of connexin expression in HCC. Thus, the title should be more precise: e.g. “…mechanistic and prognostic implications”.
Response 2: We updated the title according to your comment (Understanding the Role of Connexins in Hepatocellular Carcinoma: Molecular and Prognostic Implications).
Point 3: Tables are complex and confusing. Authors should delete the column “chemosensitivity” in table 1 and table 2 as this information are redundant to the column “clinical implications” and should delete "chemosensitivity and radiosensitivity" in table 3.
Response 3: We reviewed the tables accordingly. “Chemosensitivity and Radiosensitivity” columns’ information was deleted (and moved to “Clinical implications” columns when needed).
Point 4: Moreover, organization of the tables should be improved, as it is difficult to separate the information given in the references cited.
Response 4:
We improved the organization of our tables as follows:
-we added references after each statement
-we updated the text at a decreased-length version in some columns
-we numbered the statements at each column referring to the same reference
-we improved the lines, spaces etc., aiming to make them more readable
In the manuscript attached, the new versions of our tables are provided.
Reviewer 2 Report
Comments and Suggestions for Authors
Papadakos et al. present a narrative review manuscript of an important and impactful topic. This manuscript is very well written and manages to provide a specialized and complete vision. The figures are adequate and have a self-explanatory vision that allows the topic to be followed very correctly. The references are appropriate and current. This reviewer suggests including the role that IRS-4 has in this mechanism, due to the novel therapeutic target it may have. The authors must carry out a review of the use of English grammar. I congratulate the authors for this manuscript.
Comments on the Quality of English LanguageMinor editing of English language required.
Author Response
Dear Reviewer,
thank you for your comments, your time is highly appreciated.
Point 1: This reviewer suggests including the role that IRS-4 has in this mechanism, due to the novel therapeutic target it may have.
Response 1: We included the role of IRS-4, lines 407-412 (we added ref. 99 and updated references accordingly)
Point 2: The authors must carry out a review of the use of English grammar.
Response 2: Minor editing in English language was performed as asked (e.g., lines 66, 171, 188, 198, 221, 273, 324, 415, 425, 437, 438, 485, 502-3, 541, 621, 680, 713-4, 722), the changes are highlighted. We also edited the term of NAFLD in lines 570-1 to come in concordance with the terms referred at the introduction paragraph lines 61-2.